# PPAR Gamma Agonist Leriglitazone Recovers Alterations Due to Pank2-Deficiency in hiPS-Derived Astrocytes

**DOI:** 10.3390/pharmaceutics15010202

**Published:** 2023-01-06

**Authors:** Paolo Santambrogio, Anna Cozzi, Ivano Di Meo, Chiara Cavestro, Cristina Vergara, Laura Rodríguez-Pascau, Marc Martinell, Pilar Pizcueta, Valeria Tiranti, Sonia Levi

**Affiliations:** 1San Raffaele Scientific Institute, 20132 Milano, Italy; 2Fondazione IRCCS Istituto Neurologico Carlo Besta, 20126 Milano, Italy; 3Minoryx Therapeutics BE, S.A., 6041 Charleroi, Belgium; 4Minoryx Therapeutics S.L., 08302 Barcelona, Spain; 5School of Medicine, Vita-Salute San Raffaele University, 20132 Milano, Italy

**Keywords:** pantothenate kinase-2 associated neurodegeneration, leriglitazone, hiPS-derived astrocytes, nuerodegeneration with brain iron accumulation

## Abstract

The novel brain-penetrant peroxisome proliferator-activated receptor gamma agonist leriglitazone, previously validated for other rare neurodegenerative diseases, is a small molecule that acts as a regulator of mitochondrial function and exerts neuroprotective, anti-oxidative and anti-inflammatory effects. Herein, we tested whether leriglitazone can be effective in ameliorating the mitochondrial defects that characterize an hiPS-derived model of Pantothenate kinase-2 associated Neurodegeneration (PKAN). PKAN is caused by a genetic alteration in the mitochondrial enzyme pantothenate kinase-2, whose function is to catalyze the first reaction of the CoA biosynthetic pathway, and for which no effective cure is available. The PKAN hiPS-derived astrocytes are characterized by mitochondrial dysfunction, cytosolic iron deposition, oxidative stress and neurotoxicity. We monitored the effect of leriglitazone in comparison with CoA on hiPS-derived astrocytes from three healthy subjects and three PKAN patients. The treatment with leriglitazone did not affect the differentiation of the neuronal precursor cells into astrocytes, and it improved the viability of PKAN cells and their respiratory activity, while diminishing the iron accumulation similarly or even better than CoA. The data suggest that leriglitazone is well tolerated in this cellular model and could be considered a beneficial therapeutic approach in the treatment of PKAN.

## 1. Introduction

Neurodegeneration associated with pantothenate kinase 2 deficiency (PKAN, also named NBIA1, OMIM # 234200) is an autosomal recessive disease which belongs to the group of disorders named Neurodegeneration with Brain Iron Accumulation (NBIA) [1]. These are heterogeneous monogenic neurodegenerative disorders characterized by extrapyramidal symptoms and by common pathognomonic evidence of accumulation of local iron in the brain [2]. Although PKAN is a rare disease, it is among the most common forms of NBIA, accounting for around 50% of NBIA cases [3]. The diagnosis of PKAN is based on MRI findings and confirmed by genetic testing. In T2-weighted MRI images, it is characterized by the distinctive sign, called “eye of tiger”, depicting am image resulting from the hyperintense signal in the central area, corresponding to the globus pallidus and the hypointense areas surrounding it [4].

The age of onset is typical in early childhood, but there is also a less common form defined “atypical”, which manifests late, generally in the second or third decade of life. In the typical form, patients manifest walking disorders, such as ataxia, postural problems and muscular spasticity, which progress rapidly to more severe conditions, such as parkinsonism, dystonia, dysarthria, mental retardation, dementia, optic nerve atrophy and retinopathy [5,6]. The atypical form is characterized by a slower progression than the classical form, presenting a combination of mostly neuropsychiatric symptoms, among which obsessive–compulsive disorders, schizophrenia and depression stand out. Both phenotypes are characterized by the accumulation of cerebral iron. Despite several clinical trials [7], there is currently no effective therapy to slow down disease progression or to cure PKAN patients, and all treatments administered are aimed at relieving the symptoms, especially dystonia [6].

Pantothenate kinase 2 is encoded from the *PANK2* gene, which is located on chromosome 20p13 [8]. PANK2 is a protein of approximately 63 kDa that is part of the pantothenate kinases family (PANK1a, PANK1b, PANK3 and PANK4) and is the only one found within the mitochondria, specifically within the intermembrane space [9,10], while the other pantothenate kinases are found in the cytoplasm of the cell [11]. PANK2 is expressed in many tissues, but higher levels are found in the brain and liver [12,13]. Its enzymatic activity is essential for the biosynthesis of coenzyme A (CoA) [12]. Recently, different sources of CoA were demonstrated [14]. However, the majority of intracellular CoA is produced de novo through a highly conserved five-step biosynthetic pathway whose first step involves the PANK2-catalyzed transformation of pantothenic acid (or vitamin B5) [12]. The activity of PANK2 is regulated through inhibitory feedback modulated by CoA and acyl-CoA in order to maintain a correct CoA homeostasis [15].

Indeed, it remains unclear why the alterations in CoA synthesis induce the deposition of iron in the human brain. This is essentially because the animal models developed so far [10,16,17,18,19,20] do not fully reproduce the pathological phenotype, particularly they do not show the formation of severe brain iron deposits found in patients affected by PKAN.

Preclinical data obtained in vitro on patients’ fibroblasts [21] and human-induced pluripotent stem (hiPS)-derived glutamatergic neurons [22,23,24] indicated that they did not present iron accumulation while showing mitochondrial dysfunction, oxidative stress, reduced respiratory activity and calcium dyshomeostasis. The treatment with CoA was effective in mitigating all these pathological phenotypes. Recently, it was also demonstrated that the PKAN hiPS-derived medium spiny neurons, a population composed by GABAergic neurons and astrocytes, also showed severe cytosolic iron accumulation, mainly in astrocytes. This was in agreement with the neuropathological analysis of brain PKAN patients that showed iron overload in glia cells [25].

Also in the PKAN hiPS-derived astrocytes, CoA treatment reduced iron deposition [26]. Consequently, these data strongly support the usage of astrocytes as an in vitro platform to test compounds aimed at reverting pathological features of the human pathology. In this regard, we decided to verify if another compound, known to have an effect in improving mitochondrial function, can be effective even if not directly involved in CoA biosynthesis. We tested leriglitazone, a brain penetrant, selective agonist of Peroxisome proliferator-activated receptor gamma (PPARγ) that is being developed for the treatment of central nervous system (CNS) diseases by Minoryx Therapeutics [27].

Leriglitazone (5-[[4-[2-[5-(1-hydroxyethyl)pyridin-2-yl]ethoxy] phenyl]methyl]-1,3-thiazolidine-2,4-dione hydrochloride) is the hydrochloride salt of the active metabolite M4 (M-IV) of pioglitazone (Actos^®^, Takeda). PPARγ is a transcription factor, which, among various essential roles for cellular life, is a key regulator of mitochondrial function and biogenesis, energy metabolism, anti-oxidant defense and inflammation [28,29]. Leriglitazone showed a robust preclinical proof-of-concept in in vitro and in vivo preclinical models of multiple neurodegenerative diseases, such as X-linked adrenoleukodystrophy (X-ALD) and Friedreich’s ataxia (FRDA), by modulating pathways leading to mitochondrial dysfunction, oxidative stress, neuroinflammation, demyelination and axonal degeneration. Phase 1 data showing CNS exposure and target engagement in humans was also reported [27,30]. In addition, clinical trials in adult X-ALD (ADVANCE; NCT03231878) and FRDA (FRAMES; NCT03917225) have been completed, and a study on pediatric X-ALD patients with cerebral ALD (NEXUS) is ongoing.

Herein, we used the hiPS-derived PKAN astrocytes to monitor the effect of leriglitazone (in figures referred as MIN-102) in comparison with CoA. We noted that the treatment with leriglitazone did not affect the differentiation of neuronal precursors into astrocytes. Most importantly, it improved the viability of PKAN cells, increased the mitochondrial respiration, and, at the same time, significantly reduced iron accumulation.

## 2. Materials and Methods

### 2.1. Generation of Human Astrocytes from NPC

The hiPS clones for controls and PKAN patients were previously generated and fully characterized [22]. Subsequently, the hiPS clones were differentiated into a pure and stable population of self-renewable NPCs as in [22]. The obtained NPCs were maintained in DMEM-F12 supplemented with 2 mM L-glutamine (Sigma, St. Louis, MI, USA), 1% Pen/Strep (Sigma), B27 (1:200, Life Technologies, Carlsbad, CA, USA), N2 (1:100, Life Technologies), and βFGF (20 ng/mL, Tebu-Bio, Paris, France). When the seeded cells reached 60% confluence, heat-inactivated FBS (20%, Thermo Fisher Scientific, Waltham, MA, USA) was added to the medium, and the culture was maintained for more than 45 days in order to produce astrocytes with a good level of maturation. Once confluent, the purity of cultures was assessed.

### 2.2. Astrocytes Treatments

Astrocytes were treated with leriglitazone (MIN-102, Minoryx Therapeutics, Mataró, Spain) in the range from 3 nM to 3 μM for three days after they reached maturation to determine the optimal dose of the drug. In all the other experiments, astrocytes were differentiated in the presence or absence of 100 nM of leriglitazone or 25 μM of CoA for more than 45 days with fresh medium and drugs changed every 2–3 days.

### 2.3. Immunofluorescence

Cells grown on coverslips were fixed in 4% paraformaldehyde and processed for immunofluorescence as described in [31]. Immunofluorescence was performed using the following specific antibodies: mouse anti excitatory amino acid transporter2 (EAAT2, 1:200; SC-365634, Santa Cruz Biotchenology, Dallas, TX, USA); rabbit anti glial fibrillar acidic protein (GFAP, 1:250; GA524, Agilent Technologies, Santa Clara, CA, USA); Alexa fluor 546 donkey anti mouse IgG (1:800; IS20305, Immunological Sciences, Rome, Italy); Alexa fluor 488 goat anti rabbit IgG (1:800; IS20015, Immunological Sciences). Fluorescence images were acquired using a Zeiss Axio Observer Z1 microscope equipped with a Hamamatsu EM-CCD 9100-02 camera and Volocity acquisition software.

### 2.4. Cell Viability Assay

A total of 2 × 10^4^ astrocytes/well in 96-well plates were grown in the appropriate medium at 37 °C for 18 h and then incubated with 10 μL of MTT solution (Sigma-Aldrich 5 mg/mL in phosphate-buffered saline) for 2 h at 37 °C. The color absorbance was read at 570 nm on an iMark Microplate Reader (BioRad, Hercules, CA, USA).

### 2.5. Determination of Respiratory Activity

The oxygen-consumption rate (OCR) of astrocytes was measured with an XF96 Extracellular Flux Analyzer (Seahorse Bioscience, North Billerica, MA, USA) as previously described (Orellana).

### 2.6. Iron Staining

Astrocytes were differentiated for more than 70 days in the presence or absence of 100 nM leriglitazone or 25 μM CoA and then stained for iron content with Perls reaction by incubation for one hour in 1% potassium ferrocyanide, 1% hydrochloric acid in distilled water. Cells were counterstained with nuclear fast red (Sigma). Images were taken on Zeiss AxioImager M2m equipped with AxioCam MRc5 using 40× objective.

### 2.7. Statistical Analysis

GraphPad PRISM^®^ software (GraphPad Software Inc., San Diego, CA, USA) was used togenerate the graphs and to perform statistical analysis. The results are expressed as mean ± SD or ±SEM. The statistical analysis performed was indicated in the corresponding figure legend. Significant *p*-values: * *p* < 0.05; ** *p* <0.01; *** *p* < 0.001; **** *p* < 0.0001 were considered.

## 3. Results

### 3.1. A Short Treatment with Leriglitazone Does Not Interfere with the Viability and Respiration of Astrocytes

Starting from the previously obtained hiPS clones [22], we differentiated three neonatal normal subjects and three PKAN patients, carrying mutations causing PANK2 deficiency as described in Table 1 into hiPS-derived neuronal precursor cells (NPC) and further differentiated in astrocytes [26].

The differentiation method, previously set up in [26] and described in MM, led to a pure mature astrocyte culture in more than 45 days starting from NPCs. The final astrocyte population (d-astrocytes) was positive to GFAP- and EAAT2-specific markers (Figure 1A) without any evident morphological difference between healthy controls and PKAN patients.

To determine the optimal dose for leriglitazone treatment, we added to the growth medium increasing concentrations of the drug ranging from 3 nM to 3 μM and maintained the mature d-astrocytes culture of one control and one patient for 3 days. At the end, we analyzed the d-astrocytes’ viability and respiration, two features affected in PKAN neuronal models [22,24], to verify the safety of the drug treatment. The results showed that, although the viability of PKAN d-astrocytes were lower than the control, confirming previous results obtained in glutamatergic neurons [22], the treatment had a similar effect in both the patient and control cells and maintained a viability above 90% up to 300 nM of drug (Figure 2A). The respiratory activity of d-astrocytes was significantly lower in the patient compared with the control, but the treatment did not worsen respiration (Figure 2B).

### 3.2. An Extended Treatment with Leriglitazone Does Not Interfere with the Astrocyte Differentiation

Based on these results, we decided to use a concentration of 100 nM of MIN-102 and extend the time of treatment to assess the effect of longer treatment with the drug and directly compare the results with the already proven beneficial effect of 25 μM CoA addition [22,26,32], from the beginning of the differentiation of d-astrocytes. Thus, we performed all the additional experiments using the above-described conditions of the two compounds, starting from day 0 of d-astrocyte differentiation and analyzing them in parallel. Under these conditions, all the d-astrocytes were positive to the markers specific for the mature astrocytes and showed a morphology similar in leriglitazone-treated patients and controls (Figure 1B) and comparable to the untreated d-astrocytes (Figure 1A), demonstrating that the treatment did not interfere with the morphology during differentiation. Moreover, we checked a possible effect of leriglitazone on the PANK2 peptide. The immoblotting experiments did not reveal the presence of PANK2 peptide in untreated or treated cells (not shown), as expected for these types of mutations that caused the destabilization of the protein.

### 3.3. Treatment with Leriglitazone Improves PKAN d-Astrocytes Vitality

Maintaining these identified conditions of treatment, we investigated the ability of leriglitazone in the amelioration of the pathological phenotypes. The estimated cell vitality revealed that all the untreated PKAN d-astrocytes were less viable than controls, as expected (Figure 3A). The addition of 100 nM leriglitazone or 25 μM CoA from the beginning of differentiation significantly improved the vitality of all the patient d-astrocytes, leaving the controls unaffected (Figure 3B). Moreover, in one of the patients, the leriglitazone treatment was significantly more effective than the CoA treatment.

### 3.4. Treatment with Leriglitazone Improves PKAN d-Astrocytes Respiratory Activity

Respiratory parameters of PKAN d-astrocytes were also examined by measuring the oxygen consumption rate (OCR) and comparing it with the controls. The results showed that all patient’s cells had significantly lower values of respiratory parameters than controls, showing impaired mitochondrial functionality (Figure 4). Herein, the addition of leriglitazone or CoA also enhanced the measured parameters. In particular, leriglitazone treatment significantly increased OCR in all the PKAN d-astrocytes, while CoA treatment was significant in two patient’s d-astrocytes (Figure 4), although the third one presented a tendency towards improvement. Interestingly, in two cases, the leriglitazone was able to significantly increase the respiratory activity to a greater extent than with the CoA treatment.

### 3.5. Treatment with Leriglitazone Strongly Reduces Iron Deposition in PKAN d-Astrocytes

The peculiar feature of this PKAN astrocyte model is the presence of intracellular iron deposits [26] that are similar to those detectable in the patient’s brain tissue [25]. As previously observed [26], PKAN d-astrocytes differentiated for more than 70 days start to accumulate iron in the cytosol. Iron accumulation, revealed by Perls reaction, was virtually absent in all the controls’ d-astrocytes (iron positive cells below 5%), while it was marked in the PKAN d-astrocytes (iron positive cells above 20%). Microscopy images showed that the stain appeared with a granular pattern, characteristic of iron-overloaded cells accumulated in the cytosol (Figure 5).

Treatment with CoA was already proven effective in lowering iron aggregation [26], and herein, we provided evidence that leriglitazone was also able to ameliorate the iron accumulation in all PKAN d-astrocytes (Figure 5 and Table 2) with reductions from 78 to 91% and was even more efficient than CoA (from 70 to 83%) at preventing iron deposition in all the PKAN patients.

## 4. Discussion

Leriglitazone was recently evaluated in in vitro and in vivo preclinical models of neurodegenerative disorders and clinically tested in healthy volunteers in a phase 1 study [27,30] and currently in patients. In primary rodent neurons and astrocytes of a model simulating X-linked adrenoleukodystrophy (X-ALD), leriglitazone showed the ability to decrease oxidative stress and exerted neuroprotective effects [27]. Similar beneficial properties were also confirmed on frataxin-deficient dorsal root ganglia (DRG) neurons [30], where leriglitazone increased frataxin protein levels, improved mitochondrial function, and calcium homeostasis, reducing neurite degeneration [30]. In addition, leriglitazone showed the ability: (i) to improve motor function deficits and restore mitochondrial function and biogenesis in adrenomyeloneuropathy (AMN) and FRDA animal models [30]; (ii) to reduce inflammation and microglia activation in spinal cord tissues from AMN mouse models; (iii) to decrease the neurological symptoms in the EAE (experimental autoimmune encephalomyelitis) neuroinflammatory mouse model; (iv) to prevent endothelial damage disrupting the BBB; (v) to increase myelin debris clearance and oligodendrocyte survival and myelination promoting remyelination [27,30], and (vi) to ameloriate the iron accumulation measured by QSM (quantitative susceptibility mapping) in the dentate nucleus reported recently in double-blind, randomized controlled trial in FRDA patients (FRAMES) [33]. From these studies, it can be deduced that the beneficial properties of leriglitazone are pleiotropic through multiple pathways in CNS diseases, and in the mitochondrial diseases, the efficacy is mainly exerted by increasing biogenesis and mitochondrial functionality and reducing oxidative stress.

Herein, we checked whether this could be confirmed in our PKAN models, which are characterized by mitochondrial dysfunction. Indeed, our data indicated that the treatment of PKAN d-astrocytes during their differentiation with leriglitazone does not affect their proliferation. This could be probably explained by the recovery of the mitochondrial functionality as demonstrated by the rescue of respiratory activity. Interestingly, in two patients, the respiratory activity resulted in higher improvement after leriglitazone treatment compared to CoA treatment, suggesting a similar efficacy of the leriglitazone in rescuing mitochondrial functionality. The positive effect of the leriglitazone treatment cannot be ascribed to the increased amount of the PANK2 enzyme in these cells as seen by the absence of PANK2 peptide in untreated or treated cells. More intriguingly, the phenotype of iron deposition appeared strongly reduced after leriglitazone treatment with a slightly greater efficacy of leriglitazone as compared to CoA in reducing the percentage of cells containing iron deposits. This results is aligned with the data reported recently in FRDA patients treated with leriglitazone in the FRAMES clinical trial, showing that iron accumulation in the dentate nucleus, as assessed by QSM, was greater in the placebo arm [33]. We cannot exclude that the difference in efficacy between the two treatments is due to the greater cell permeability of leriglitazone compared to CoA. CoA is a charged molecule, and so it is not a membrane-permeable molecule, and specific transporters for CoA/dephospho-CoA have been identified [34]. However, several studies have demonstrated that the addition of CoA to the growth medium of cells had a positive effect in rescuing the pathogenic phenotype [18,21,22,24,26,32].

CoA is a key metabolic cofactor for many biochemical functions; among them, it is a vehicle of the acyl group, and it is able to activate the carbonyl groups in many reactions taking place within cells, including the tricarboxylic acid cycle and metabolism of fatty acids [35]. Furthermore, leriglitazone, by enhancing PPARγ activity, could act as a regulator of fatty acid β-oxidation and improve the energetic status of cells [36]. The evidence that leriglitazone treatment has an effect comparable to that of CoA suggests that the two compounds act on pathological mechanisms that share common alterations. Considering these details and that apparently there is not a direct involvement role of CoA/acetyl-CoA in mitochondrial iron homeostasis, we can argue a hypothesis regarding the mystery of iron deposition in PKAN pathogenesis. Cytosolic iron accumulation might be due to a non-use of iron in the mitochondria, which normally use iron for the biosynthesis of Fe-S and heme [37]. This iron unavailability might be caused by (i) iron shortage in mitochondria due to alteration of mitochondrial iron delivery, (ii) deficit of the energy needed for its utilization, (iii) a general impairment of mitochondrial function that might also involve in an unspecific manner iron-dependent biosynthesis and (iv) the concurrence of all these events. The positive effects of leriglitazone in substantially abolishing this phenotype suggest that the restoration of mitochondrial functionality, even in the absence of CoA addition, would be sufficient to avoid iron deposition. Another important factor to take into account is the ability of leriglitazone, as a PPARγ agonist, to modulate the nuclear factor-κB (NF-kB)-associated inflammatory mechanisms [38]. It is well-known that iron overload triggers inflammatory responses largely through the activation of the oxidative stress-responsive transcription NF-kB [39]. Thus, the effect of leriglitazone could synergize at various cellular levels. The proposed mechanism of action of leriglitazone in PKAN2 deficiency is depicted in Figure 6.

Further studies will be needed to define this point; however, these results are interesting from a therapeutic point of view. Until now, PKAN patients have been treated with drugs that alleviate their symptoms. These therapies are based on the usage of dopaminergic drugs, anticholinergics, tetrabenazine, baclofen and botulinum toxin. Other current therapies include the administration of iron chelating agents, such as deferiprone. In recent clinical trials, deferiprone’s long-term effect has been confirmed by the reduction of iron accumulation in the globus pallidus and a stabilization of motor symptoms [40,41,42]. Other experimental approaches for the CoA-related NBIA are based on the administration of the molecule itself or of its precursors. The administration of pantethine, a precursor of CoA, led to an improvement of the phenotype in mouse and drosophila models but did not have the same effect in humans because it is an unstable compound in serum [43]. For this reason, the use of more stable 4′-phosphopantethine and its acetylated form has been suggested and are being tested in clinical trials [7,44,45]. Another innovative therapeutic approach involves the activation of the other isoforms of pantothenate kinase to compensate for the lack of PANK2 activity, using a molecule capable of permeating the blood–brain barrier like pantazine [46]. However, its effectiveness has been proven up to now only in mouse models [46]. Thus, it is urgent to find other therapeutic approaches that efficiently cross the BBB and can be translated to the clinic. Our data shows that leriglitazone reduced iron accumulation, improved viability, and enhanced respiratory activity in an hiPS-derived astrocyte model of PKAN that recapitulates some of the main features of the human phenotype, suggesting that the treatment with leriglitazone could be further explored as a promising therapeutic approach in PKAN pathogenesis.

## 5. Patents

C.V., L.R.-P., M.M. and P.P. are employees of Minoryx and have stock options and/or patents with Minoryx.

## Figures and Tables

**Figure 1 pharmaceutics-15-00202-f001:**
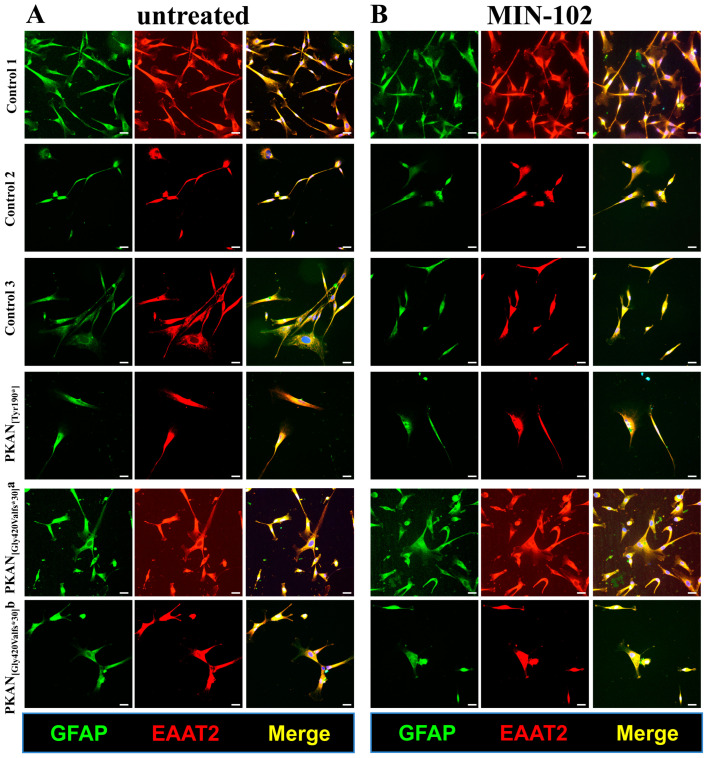
Characterization of astrocytes (50–60 days of differentiation). Controls and PKAN patients’ astrocytes were differentiated from NPC by growing them in astrocytes’ medium in the presence (**B**) or absence (**A**) of MIN-102 (100 nM). Cells were then fixed and stained with the astrocyte markers GFAP (green) and EAAT2 (red) to identify mature astrocytes. Nuclei were stained with Hoechst (blue). Images were taken on Zeiss Axio Observer.Z1 equipped with Hamamatsu 9100-02 EM CCD camera using 20× objective. Scale bar = 20 μm.

**Figure 2 pharmaceutics-15-00202-f002:**
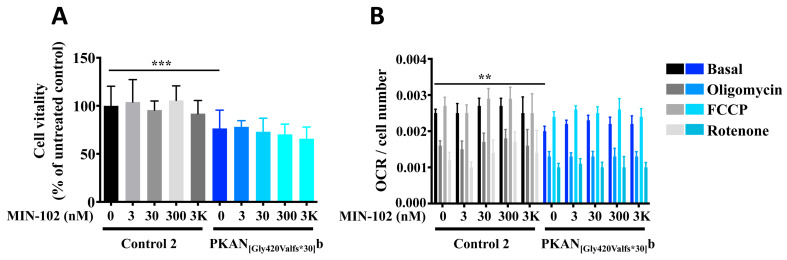
Plots showing the vitality and the oxygen consumption rate (OCR) measurements on the astrocytes (46 days of differentiation). Control (Control 2) and PKAN patient (PKAN_[Gly420Valfs*30]_b) astrocytes were incubated in the presence or absence of MIN-102 for the last three days of differentiation. (**A**) Cells were incubated with MTT solution, and color absorbance at 570 nm was measured. Data are expressed as % relative to the untreated control astrocytes. One-way ANOVA, *** *p* < 0.001. Mean ± SD, sixteen replicates for the untreated samples and eight replicates for the MIN-102 treated. (**B**) Oxygen consumption rate (OCR) was measured with XF96 Extracellular Flux Analyzer (Seahorse). Plot shows OCR normalized on cell numbers. Two-way ANOVA, ** *p* < 0.01. Mean ± SEM, fourteen replicates for the untreated samples and eight replicates for the MIN-102 treated.

**Figure 3 pharmaceutics-15-00202-f003:**
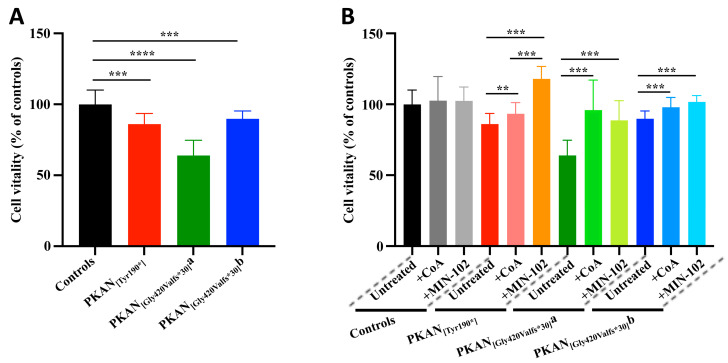
Plots showing the vitality of the astrocytes (50–60 days of differentiation). Cells were grown in astrocyte medium in the presence or absence of MIN-102 (100 nM) or CoA (25 μM) from day 0 of differentiation, and vitality was measured by MTT assay. (**A**) Plot showing the untreated astrocytes only, and (**B**) plot showing the complete experiment. All data are expressed relative to the untreated control astrocytes. One-way ANOVA, ** *p* < 0.01, *** *p* < 0.001, **** *p* < 0.0001. Mean ± SD, at least sixteen replicates for all the samples.

**Figure 4 pharmaceutics-15-00202-f004:**
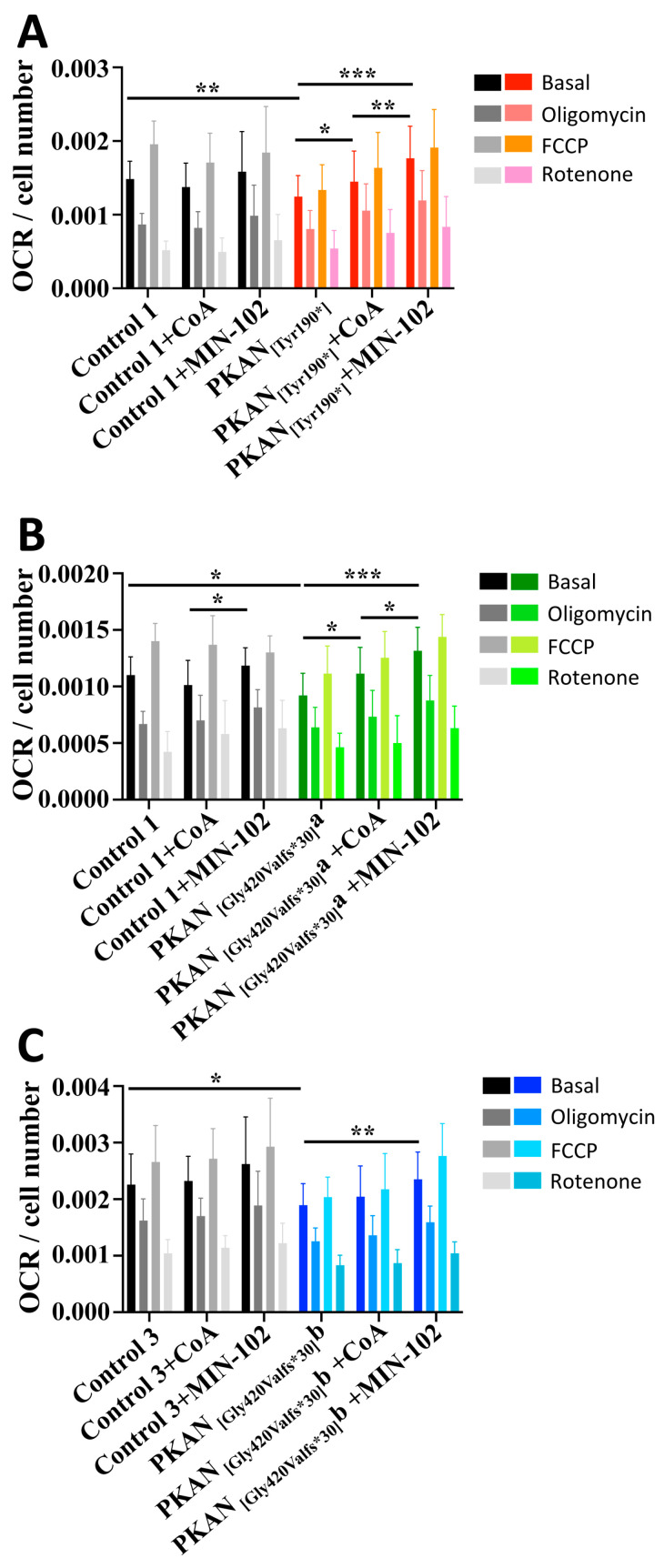
Oxygen consumption rate (OCR) measurements on astrocytes (50–60 days of differentiation). Controls and PKAN patients’ astrocytes were grown in the presence or absence of 100 nM MIN-102 or 25 μM CoA from day 0 of differentiation. Oxygen consumption rate was measured with XF96 Extracellular Flux Analyzer. (**A**–**C**) Plots show OCR normalized on cells number (subject names and conditions are detailed in the figure labels). Unpaired *t* test, * *p* < 0.05, ** *p* < 0.01, *** *p* < 0.001. Mean ± SEM, at least thirteen replicates per sample.

**Figure 5 pharmaceutics-15-00202-f005:**
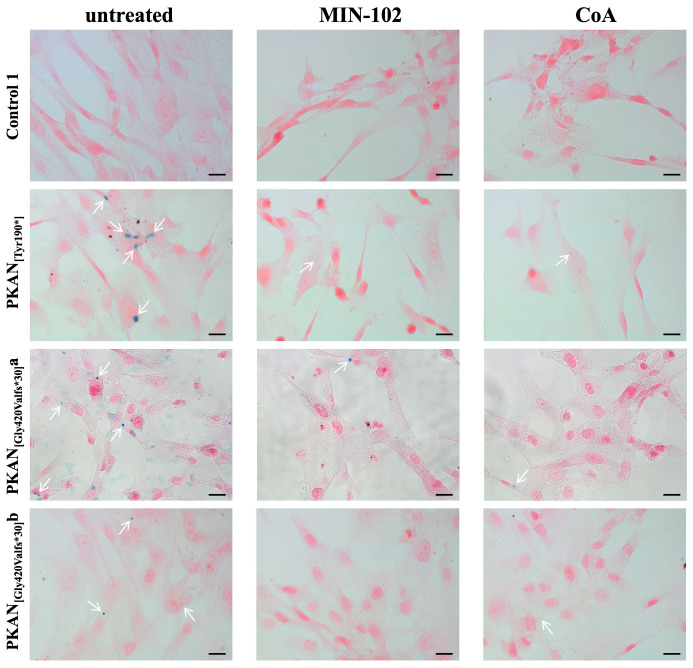
Iron accumulation in astrocytes (80 days of differentiation). Astrocytes (Control 1 and PKAN patients) were differentiated in the presence or absence of 100 nM MIN-102 or 25 μM CoA. Fixed astrocytes were stained with Perls reaction to detect iron granules (blue) and counterstained with nuclear fast red. Images were taken on Zeiss AxioImager M2m equipped with AxioCam MRc5 using 40× objective. White arrows indicate iron granules. Scale bar = 20 μm.

**Figure 6 pharmaceutics-15-00202-f006:**
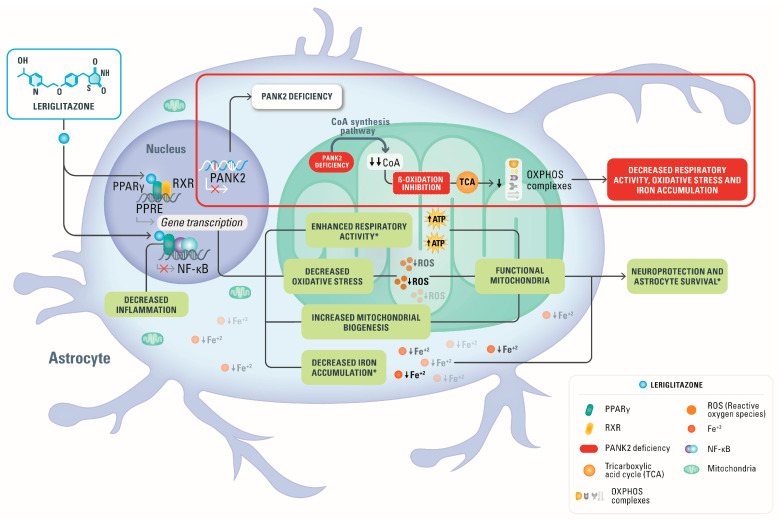
Proposed mechanism of action of leriglitazone. The beneficial effects of leriglitazone that have been confirmed in the current study on the hiPS-derived astrocytes from PKAN patients are indicated with an asterisk *: leriglitazone enhanced respiratory activity, decreased iron accumulation and increased astrocyte survival. Moreover, leriglitazone decreased oxidative stress, increased mitochondrial biogenesis and decreased inflammation as reported in other preclinical studies. All these effects could lead to an improvement of mitochondrial function resulting in neuroprotection and increased astrocyte survival.

**Table 1 pharmaceutics-15-00202-t001:** Samples utilized.

Subject (Code)	DNA Mutation	Protein Mutation
NeoL (Control 1)	---	---
CB (Control 2)	---	---
DIGI (Control 3)	---	---
Patient 1 (PKAN_[Tyr190*]_)	c. [569_570insA]	p. [Tyr190*]
Patient 2 (PKAN_[Gly420Valfs*30]_a)	c. [1259delG]	p. [Gly420Valfs*30]a
Patient 3 (PKAN_[Gly420Valfs*30]_b)	c. [1259delG]	p. [Gly420Valfs*30]b

**Table 2 pharmaceutics-15-00202-t002:** Number of total and iron-positive astrocytes from experiment shown in Figure 5.

Sample	Counted Fields	Total Cells	Fe Positive Cells	Fe Positive Cells (%)	Reduction of Fe %
Control 1 ut	10	962	45	4.7	---
Control 1 MIN-102	10	308	4	1.3	74.5
Control 1 CoA	9	323	7	2.2	73.5
Control 2 ut	10	71	2	2.8	---
Control 2 MIN-102	10	31	0	0	100
Control 2 CoA	3	22	0	0	100
Control 3 ut	3	77	1	1.3	---
Control 3 MIN-102	3	43	0	0	100
Control 3 CoA	3	44	0	0	100
PKAN_[Tyr190*]_ ut	6	104	20	19.2	---
PKAN_[Tyr190*]_ MIN-102	6	111	5	4.5	78.4
PKAN_[Tyr190*]_ CoA	6	98	6	6.1	70.7
PKAN_[Gly420Valfs*30]_a ut	10	556	127	22.8	---
PKAN_[Gly420Valfs*30]_a MIN-102	10	708	18	2.5	89
PKAN_[Gly420Valfs*30]_a CoA	10	152	10	6.6	71
PKAN_[Gly420Valfs*30]_b ut	10	148	39	26.4	---
PKAN_[Gly420Valfs*30]_b MIN-102	10	163	4	2.4	90.9
PKAN_[Gly420Valfs*30]_b CoA	7	109	5	4.6	82.6

ut = untreated cells; MIN-102 = 100 nM MIN-102 treated cells; CoA = 25 μM CoA treated cells.

## Data Availability

All data are available upon reasonable request.

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
