# Peer review of "PPAR Gamma Agonist Leriglitazone Recovers Alterations Due to Pank2-Deficiency in hiPS-Derived Astrocytes"

_pharmaceutics, 2023, doi:10.3390/pharmaceutics15010202_

Round 1

Reviewer 1 Report

The PPARgamma agonist leriglitozone has been shown effective in treating several models of neurodegenerative diseases. Here the authors adopted this compound in evaluating its efficacy in hiPS-derived PKAN patient astrocytes. They found leriglitozone had about equivalent or slightly better rescues in restoring mitochondrial respiration, reducing iron accumulation and regaining vitality for these mutant astrocytes.  The experimental design is straightforward, and result clean. I have only the following comments.

1. Why not use neuron (instead of astrocyte) here in the work?

2. The key molecular defect of PANK2 mutation should be lack of CoA. Does leriglitozone improve CoA levels, and if not, does combination of leriglitozone and CoA achieve better results? Understanding this is not only important in providing mechanistic insight in the rescue, but also helpful in improving the therapy design. 

Author Response

We thank the Reviewer 1 for the positive evaluation of our work and suggestions.

  1. We added into the introduction a deeper explanation of the difference between neurons and astrocytes phenotypes pointing out that the iron accumulation is present mainly in astrocytes (Ref. 26), which has been demonstrated to be the iron positive cell type in human PKAN brains (Ref. 25). The text has been changed as the following:

Preclinical data obtained in vitro on patients’ fibroblasts [21] and human-induced pluripotent stem (hiPS)-derived glutamatergic neurons [22][23][24] indicated that they did not present iron accumulation while showing mitochondrial dysfunction, oxidative stress, reduced respiratory activity, and calcium dyshomeostasis. The treatment with CoA was effective in mitigating all these pathological phenotypes. Recently, it was also demonstrated that the PKAN hiPS-derived medium spiny neurons, a population composed by GABAergic neurons and astrocytes, showed also severe cytosolic iron accumulation mainly in astrocytes. This was in agreement to the neuropathological analysis of brain PKAN patients that evidenced iron overload in glia cells [25].”

  1. We agree that that the measure of the CoA level is important and that the effect of leriglitazone might have a better effect in combination with CoA. However, we have not measured the CoA level in our cellular model and did not perform experiments with the combination of the two drugs, since the main aim of this work was to evaluate, for the first time, the efficacy of this drug in PKAN model and to compare its effects with the already used CoA.

Reviewer 2 Report

The authors have designed the study to evaluate the effects of leriglitazone on PKAN-human iPSC derived astrocytes versus control human iPSC derived astrocytes.

The study highlights the mitigating effects of leriglitazone on iron deposition, astrocyte viability and improved respiratory activity. The authors have also discussed the limitations of their study such as: 

1. Indirect link between CoA biosynthetic pathway and brain Iron deposition as observed in patient MRI scans.

2. Requirement of further studies to clearly elucidate the underlying mechanisms of protective effects exerted by leriglitazone.

The study highlights the urgency of drug development for rare neurological diseases. I recommend the manuscript for publication with minor revisions.

The comments are as follows:

1.  It is important to mention which cell population in the brain show iron deposition in PKAN patients.

2. Does leriglitazone have any cell specific effects? Will the effects on astrocytes be any different than on neurons? Maybe effects of  leriglitazone on an organoid model be elucidative of its mechanisms?

3. Is there a specific reason why the effects of leriglitazone treatment was studied in astrocytes but not on human iPSC derived neurons? 

3. Quality of images are poor. Better resolution needed for publication purposes.

Author Response

We thank the Reviewer 2 for the positive evaluation of our work and suggestions.

  1. We added the information about the cell type positive for iron in human PKAN brain in the introduction. The text has been changed as following:

“…..This was in agreement to the neuropathological analysis of brain PKAN patients that evidenced iron overload in glia cells [25].”

  1. The previous published data obtained using this drug on other disease models (cited in introduction and discussion) did not show cell specificity effect. Thus, we do not expect difference in efficacy between treatment on astrocytes or neurons.

We agree that using organoid model will be more elucidative and we programmed this for future work.

  1. We added into the introduction a deeper explanation of the difference between neurons and astrocytes phenotypes pointing out that the iron accumulation is present mainly in astrocytes (Ref. 26), which has been demonstrated to be the iron positive cell type in human PKAN brains (Ref. 25). The text has been changed as the following:

Preclinical data obtained in vitro on patients’ fibroblasts [21] and human-induced pluripotent stem (hiPS)-derived glutamatergic neurons [22][23][24] indicated that they did not present iron accumulation while showing mitochondrial dysfunction, oxidative stress, reduced respiratory activity, and calcium dyshomeostasis. The treatment with CoA was effective in mitigating all these pathological phenotypes. Recently, it was also demonstrated that the PKAN hiPS-derived medium spiny neurons, a population composed by GABAergic neurons and astrocytes, showed also severe cytosolic iron accumulation mainly in astrocytes. This was in agreement to the neuropathological analysis of brain PKAN patients that evidenced iron overload in glia cells [25].”

  1. We added the high-quality pictures in the final version.

Reviewer 3 Report

The purpose of this study was to investigate whether the PPARgamma antagonist leriglitazone can effectively ameliorate mitochondrial defects in an in vitro model of Pantothenate kinase-2-associated neurodegeneration (PKAN). While the idea of the study seems interesting, the methodology of this study is very limited and suffers from many shortcomings that, in my opinion, disqualify this work from publication:

1) The major problem is that this work does not present the molecular mechanisms underlying the effect of leriglitazone on mitochondria function in the PKAN model.

2) Why were hiPS differentiated into astrocytes and not neurons? What is the reason for using astrocytes in studying the protective effects of leriglitazone?

3) The treatment paradigm of leriglitazone is also not clear. Was it added daily or at each change of medium? If so, how often (e.g., twice a week)? Was it added in the same manner as CoA?

4) The MTT test is not sufficient to detect cell death because it could also be influenced by higher cell proliferation. Additional tests (e.g., LDH) should be performed.

5) Immunocytochemistry is insufficient to prove the drug's negligible effect on the astrocyte differentiation process.

6) Why was a leriglitazone concentration of 100 nM used for the study, since higher drug concentrations should have a negligible effect on cell viability?

7) Why were particular experiments performed on different days of astrocyte differentiation?

Author Response

We thank the Reviewer 3 for the positive evaluation of our work and suggestions.

  1. The main aim of this work was to evaluate the efficacy of leriglitazone in PKAN patients’ derived cells and to compare its effects with the already used CoA, therefore elucidation of its mechanism is behind the scope of this work.
  2. The previous published data obtained using this drug on other disease models (cited in introduction and discussion) did not show cell specificity effect. Thus, we do not expect difference in efficacy between treatment on astrocytes or neurons.

We added into the introduction a deeper explanation of the difference between neurons and astrocytes phenotypes pointing out that the iron accumulation is present mainly in astrocytes (Ref. 26), which has been demonstrated to be the iron positive cell type in human PKAN brains (Ref. 25). The text has been changed as the following:

Preclinical data obtained in vitro on patients’ fibroblasts [21] and human-induced pluripotent stem (hiPS)-derived glutamatergic neurons [22][23][24] indicated that they did not present iron accumulation while showing mitochondrial dysfunction, oxidative stress, reduced respiratory activity, and calcium dyshomeostasis. The treatment with CoA was effective in mitigating all these pathological phenotypes. Recently, it was also demonstrated that the PKAN hiPS-derived medium spiny neurons, a population composed by GABAergic neurons and astrocytes, showed also severe cytosolic iron accumulation mainly in astrocytes. This was in agreement to the neuropathological analysis of brain PKAN patients that evidenced iron overload in glia cells [25].”

  1. The modality of leriglitazone treatment has been detailed in Material and Method as following:

“In all the other experiments, astrocytes were differentiated in the presence or absence of 100nM of leriglitazone or 25mM of CoA for more than 45 days with fresh medium and drugs changed every 2-3 days.”

  1. We use MTT assay to check difference in viability between treated and untreated cells. In control cells we have no evidence that the treatments have any effect on the viability of the cells. Thus, we interpreted the positive effect on patient cells as an improvement of cell vitality that might be due to both the vitality and proliferation enhancement.
  1. We characterized the astrocytes by the expression specific markers for mature astrocytes ( EEAT 2 and GFAP) and both control and PKAN d-astrocytes resulted positively stained. No difference in the expression of these two markers between untreated and treated cells emerged. Since the drug was maintained during differentiation, we concluded that it did not have any effect on maturation.
  2. We think that the use of the minimum effective dose of drug showing a positive effect is the right choice.
  3. The variation in time points in which the experiments have been performed are due to the difference in the presentation of the specific phenotype evaluated. For example: the respiration defects are already present after about 50 days of differentiation, while the iron deposition needed more than 70 days to appear.

Round 2

Reviewer 3 Report

Revised paper has been improved and now ready for publication